# Relationship between Blood Vessels and Migration of Neuroblasts in the Olfactory Neurogenic Region of the Rodent Brain

**DOI:** 10.3390/ijms222111506

**Published:** 2021-10-25

**Authors:** Marcela Martončíková, Anna Alexovič Matiašová, Juraj Ševc, Enikő Račeková

**Affiliations:** 1Department of Regenerative Medicine and Cell Therapy, Institute of Neurobiology, Biomedical Research Center, Slovak Academy of Sciences, Šoltésovej 4, 040 01 Košice, Slovakia; racekova@saske.sk; 2Department of Cell Biology, Institute of Biology and Ecology, Faculty of Science, Pavol Jozef Šafárik University in Košice, Šrobárova 2, 041 54 Košice, Slovakia; anna.alexovic.matiasova@upjs.sk (A.A.M.); juraj.sevc@upjs.sk (J.Š.)

**Keywords:** blood vessels, neuroblast migration, rostral migratory stream, adult neurogenesis

## Abstract

Neural precursors originating in the subventricular zone (SVZ), the largest neurogenic region of the adult brain, migrate several millimeters along a restricted migratory pathway, the rostral migratory stream (RMS), toward the olfactory bulb (OB), where they differentiate into interneurons and integrate into the local neuronal circuits. Migration of SVZ-derived neuroblasts in the adult brain differs in many aspects from that in the embryonic period. Unlike in that period, postnatally-generated neuroblasts in the SVZ are able to divide during migration along the RMS, as well as they migrate independently of radial glia. The homophilic mode of migration, i.e., using each other to move, is typical for neuroblast movement in the RMS. In addition, it has recently been demonstrated that specifically-arranged blood vessels navigate SVZ-derived neuroblasts to the OB and provide signals which promote migration. Here we review the development of vasculature in the presumptive neurogenic region of the rodent brain during the embryonic period as well as the development of the vascular scaffold guiding neuroblast migration in the postnatal period, and the significance of blood vessel reorganization during the early postnatal period for proper migration of RMS neuroblasts in adulthood.

## 1. Introduction

Postnatal neurogenesis is restricted to two sites in the mammalian brain, the subventricular zone (SVZ) of the lateral ventricles and the subgranular zone (SGZ) of the dentate gyrus of the hippocampus. Unlike newborn cells in the SGZ destined for the overlying granule cell layer [1], the cells arising in the SVZ have to migrate a relatively long way through the rostral migratory stream (RMS) to their final destination, the olfactory bulb (OB). 

In the SVZ, the largest neurogenic zone of the adult brain, astrocyte-like cells which are neural stem cells (type B cells) [2] give rise upon activation to transit-amplifying cells (type C cells), which generate neuroblasts (type A cells) and glia [3,4]. Neuroblasts originating in the SVZ of rodents migrate tangentially in chains several millimeters along the RMS. The chains of neuroblasts are surrounded by a meshwork of astrocytes. Once the neuroblasts reach the OB, they detach from the chains and migrate radially to the bulbar layers, where they differentiate into interneurons [5]. The migration of neuroblasts in the migratory pathway surrounded by mature tissue is a complex process. The mechanism of migration of postnatally-generated neural precursors differs from that in prenatal development. In contrast to the embryonic period, SVZ-derived neuroblasts in the adult brain migrate along the RMS without the aid of radial glia processes, since the radial glia disappear and/or transform into astrocytes after birth [6]. Instead, neuroblasts use each other as a physical substrate, forming chains of cells based on cell-cell contacts in a process termed homophilic migration. This type of neuroblast movement is faster than other types of migration [7,8]. In addition, postnatally-generated neuroblasts are able to divide while migrating through the RMS [9,10,11], unlike in the embryonic period when only postmitotic neuroblasts migrate to the final position [12]. Recently, growing body of evidence indicates that blood vessels in the adult brain play a role in guiding neuroblast migration in the RMS, as well as, providing molecular cues affecting migration. It has been shown that vasculature is a prominent feature of stem-cell niches and plays an important role in their regulation and maintenance [13,14]. The vasculature of the main neurogenic regions (SGZ and SVZ) is highly organized compared to that in non-neurogenic brain areas [13,14,15,16,17,18], and is characterized by a dense network of blood vessels which provide a substrate for progenitor cells [13,14,15,19]. Blood vessels in the RMS serve as a migration-promoting scaffold due to their specific arrangement. It has been shown that, for the proper migration of neuroblasts in the SVZ-RMS-OB neurogenic region, what is essential is not only the establishment of vessels in the embryonic period, but also their reorganization into a migration-promoting scaffold during the early postnatal period [20,21].

This review focuses on the role of blood vessels in relation to neuroblast migration in the SVZ-RMS-OB, the development and vascularization of the presumptive neurogenic region during the embryonic period, the relevance of blood vessel rearrangement in the RMS during the early postnatal development, and the function of blood vessels in neuroblast migration in the neurogenic area of the adult brain.

## 2. Development and Vascularization of the Telencephalon and the Rostral Migratory Stream during the Embryogenesis of Rodents

To provide a comprehensive view of the role of blood vessels in neuroblast migration in adult RMS, it is important to address the development of the telencephalon with emphasis on the presumptive rostral migratory stream and its vascularization during the embryonic and perinatal period For more detail, the timing of developmental events for mice and rats is summarized in Table 1 and vascularization of these regions is summarized in Table 2.

### 2.1. Development of the Telencephalon and the RMS

The CNS of rodents starts to develop during the early period of embryogenesis at the end of the first third of development in the region of the dorsal midline ectoderm. First, neuroepithelial cells establish the neural plate, which subsequently transforms into the neural tube (Table 1) [22,24] in the process known as neurulation [23]. During this period and later, the neural tube is composed of actively-proliferating neuroepithelial cells, which play a fundamental role in the formation of CNS tissue [45,46]. The anterior part of the neural plate gives rise to the prosencephalon, which subsequently subdivides into the diencephalon and the telencephalon. The telencephalon derives from cells at the rostral margin of the neural plate [47,48]. Subsequently, the developing telencephalon subdivides into ventral (subpallial) and dorsal (pallial) territories. The embryonic dorsal telencephalon can be divided into two main regions: anterior and lateral regions, which give rise to the neocortex, and posterior and medial areas, which give rise to the hippocampus, the cortical hem, and the choroid plexus. The ventral telencephalon divides into three main regions: a medial domain known as the medial ganglionic eminence (MGE), and two posterior and lateral regions designated as the lateral ganglionic eminence (LGE) and the caudal ganglionic eminence (reviewed in [49,50]).

After the establishment of the neural tube in the processes of neurulation, generation of neurons, also known as neurogenesis, takes place by proliferative activity of neuroepithelial cells. These progenitors residing in the area of the telencephalic germinal zones give rise to various neuronal populations employing different modes of migration toward their target destinations, including the olfactory bulb (OB) [27]. Different types of neurons colonize the OB in specific time-windows throughout embryonic and postnatal life (Table 1) [29,36,41]. The primordium of the OB emerges in the rodent brain after the fibers of the olfactory nerve reach the anterior pole of the telencephalon (E11–E13; E–embryonic day) [27,30,31,51]. It is assumed that the OB develops primarily from the rostral part of the ventral pallium in the murine brain [52]. In this period, early neuroblasts destined for the OB differentiate into populations of mitral cells and tufted cells, expressing markers of the projection neurons such as Tbr-1, Id-2, Reelin, neurotensin, and neuropilin [33,40] and migrate radially along the basal processes of the radial glial cells toward the OB intermediate zone [33,34]. It seems that the populations of OB neurons originate in different places. Bayer and Altman [32] suggested that mitral cells of the main OB in rats originate in the septal/accumbal neuroepithelium and migrate toward the prospective OB, while tufted cells and the mitral cells of the accessory OB originate in the neuroepithelium of the OB primordium. The populations of OB interneurons originate from the proliferating subpallial progenitors of the LGE [32] and acquire selective migratory capacity toward the OB in the later periods of embryonic development [32,52]. An in vitro study indicates that the population of neuronal cells originating in the rostral LGE migrates radially toward the piriform cortex, then the cells reorient their processes and migrate tangentially toward the prospective OB. This population does not contribute to the populations of OB interneurons, but rather to the neuronal populations expressing markers of relay neurons such as Tbr-1 [38]. Moreover, OB interneurons derived from the precursors of the embryonic LGE in the later periods of development retain their restricted migratory capacity toward the OB, even if grafted into the SVZ of adult mice [53], since E14, TUJ1, or MAP2 positive neuroblasts organize themselves into chains leading from the ventral anterior tip of the lateral ventricles toward the OB ventricles. Migrating neuronal cells in the RMS have bipolar morphology and express migratory markers such as NCAM or PSA-NCAM. As the development proceeds, neuronal cells in the RMS organize themselves into a dense patch (by E16), which increases in size and elongates toward the OB. Cells surrounding the patch, also termed non-patch cells, remain undifferentiated until birth, when they start to express neuronal markers. During migration in the embryonic RMS, patch cells forming its core start to express markers of GABAergic interneurons such as GAD65 or GAD67 [37]. After reaching the OB, neuronal cells in the embryonic and early postnatal RMS migrate along the processes of radial glial cells [7,43] to their final position and differentiate into populations of OB interneurons such as periglomerular or granule cells [36,37,41] expressing GABA, calretinin, calbindin, GAD65, GAD67, or tyrosine hydroxylase [29,37]. Astrocytes emerge in the developing RMS (in the region of the patch) of rats during the late embryonic periods (at E16) [37], however their number increases especially during the first postnatal weeks [44,54].

### 2.2. Vascularization of the Telencephalon and the RMS during the Embryonic Period

Development of the CNS from the very early beginning is inextricably linked with the formation of the vasculature in the processes of vasculogenesis. As soon as the neurulation of the rodent embryo begins, vascularization of the CNS is initiated in the mesoderm surrounding the neural tube [55] (Table 2). First, establishment of external vasculature around the developing neural tube of rodents occurs through the coalescence of endothelial cells differentiating from angioblasts into the primitive sinusoid vessels known as the perineural vascular plexus (PNVP) (Table 2) [55,56,57]. PNVP formation is controlled by vascular endothelial growth factor (VEGF), released by the neuroepithelium, which stimulates blood vessel growth by binding to the VEGF receptors expressed on the endothelial cells [58,59]. The PNVP spreads along the rostro-caudal axis of the neural tube and supplies the neuroepithelium with oxygen and nutrients [25,60,61]. As development proceeds, the rostral neural tube grows and undergoes segmentation into the primary and secondary brain vesicles, giving rise to the telencephalon [22,24,27]. Subsequently, internal vascularization of the telencephalon is established through the mechanisms of sprouting angiogenesis driven by the increased metabolic demands of neuronal progenitors during the onset of neurogenesis [25,60,62]. Blood vessels ingress the telencephalon from the PNVP and from the basal artery, located on the floor of the basal ganglia primordium, giving rise to the periventricular vascular plexus (PVVP) (Table 2) [63,64]. During the early phases of angiogenesis, blood vessels in the mouse telencephalon extend, sprout, and form new branches (E14.5–E16.5), while in later periods, these branches fuse together and retract (E16.5–E18.5) [65]. Development of the PVVP in the mouse telencephalon progresses in ventral-to-dorsal direction, since the PVVP emerges first in the subpallial preoptic area, continuing toward the MGE, LGE, ventral pallium, and to the dorsal pallial regions, successively [60,63]. It was found that the ventral-to-dorsal pattern of PVVP growth follows the expression pattern of the region-specific transcription factors expressed by neuronal progenitors such as Nkx2.1, Dlx1, and Dlx2 observed in the endothelium of the ventral PVVP, or Pax6 expressed by the PVVP endothelium in the dorsal telencephalon [60,63,66]. 

The studies examining structural features and arrangement of blood vessels in the presumptive forebrain neurogenic region of mice during development revealed that blood vessels re-orient during the early embryonic stages and become more complex as development proceeds [67,68]. At E14.5, radially-oriented blood vessels predominate in this region. They are short, straight, and relatively unbranched. A few longer blood vessels border the RMS elbow [67]. At E16, tangentially-oriented blood vessels start to appear in the presumptive RMS region [68]. From E17.5 until postnatal day 1 (P1), blood vessels within the forebrain neurogenic niche undergo gradual remodeling. At E17.5, blood vessels follow the longitudinal axis of the primordial forebrain neurogenic region, with the exception of blood vessels located in the olfactory placode, which form loop-like structures. At this age, blood vessels become longer, branched, and more frequent along the borders of the RMS [67].

Finally, intense vascular remodeling continues during perinatal and early postnatal ages in the forebrain neurogenic region [20,67,68]. At early postnatal stages (during the first postnatal week), blood vessels become more aligned in the longitudinal direction of the RMS and parallel to each other [68]. During the later postnatal stages, the blood vessels continue increasing in complexity and length [67].

## 3. Angiogenesis in the Brain during the Postnatal Period and Mutual Coordination between Nervous and Vascular Systems

In the postnatal period, angiogenesis in the brain gradually attenuates. In the rat brain, angiogenesis is complete within approximately three weeks after birth [69,70]. Then, proliferation of endothelial cells is markedly down-regulated and angiogenesis is linked only with vascular growth, matching the growth of the brain [71]. On the other hand, endothelial cells, though relatively quiescent in the adult brain, can proliferate under pathological conditions such as hypoxia, tumor growth, or brain injury [70,72]. Under physiological conditions, angiogenesis persists only in restricted areas of the adult brain with continuing neurogenesis [73,74], i.e., in the SVZ of the lateral wall of the lateral ventricles and the SGZ of the dentate gyrus of the hippocampus. In these areas, neural stem cells and endothelial cells/capillaries are in close proximity within a so-called “vascular niche” [75,76].

Coordinated interactions between nervous and vascular systems have been found not only during embryogenesis but also in the neurogenic regions of the adult brain [77]. Both these systems share molecular cues, which regulate their development, function, and maintenance, such as VEGF [59,78], brain-derived neurotrophic factor (BDNF) [79], basic fibroblast growth factor (bFGF) [80,81], insulin-like growth factor-1 (IGF-1) [82], erythropoietin [83], angiopoietin [84,85], and others (for review see [75,77]), so that postnatal angiogenesis and neurogenesis are regulated by growth factors produced by both, endothelial cells and neurons [75]. 

## 4. Blood Vessels and Neuroblast Migration in Neurogenic Areas in the Postnatal Period

In general, blood vessels in the postnatal brain are responsible for supplying oxygen and nutrients and removing metabolic waste products. In addition, blood vessels together with astrocytes and pericytes constitute a blood-brain barrier which regulates transport of substances toward the brain by allowing the movement of vital substances but restricting the flow of harmful agents from blood to brain. In addition to common functions, blood vessels have been found to play another role in neurogenic areas of the adult brain. Blood vessels are an integral component of these regions and they release cues providing a microenvironment to maintain the neural stem cell niche [86]. Dividing neural precursors are in fact closely associated with blood vessels in the SGZ [74], the SVZ [13] and the RMS [68]. Moreover, a growing body of evidence shows that blood vessels in the SVZ-RMS-OB promote migration of neuroblasts from the SVZ to their target structure, the OB [16,17,21,87].

### 4.1. Migration of Neuroblasts in the RMS during the Postnatal Period

As the regulation of migration is a critical step in the integration of new neurons into the adult neuronal circuitry, conditions for migration in the RMS, which is surrounded with mature brain tissue along its entire length, must be effectively ensured. Unlike postmitotic CNS neurons migrating along radial glial processes to the cortex of the developing brain [88,89], in the adult brain RMS, SVZ-derived neuroblasts migrate independently of radial glia. They migrate using each other via cell–cell contacts, creating “chains”, and this homophilic mode of cell movement is also called chain migration [44,90]. The migration of neuroblasts through the RMS is affected by both chemoattractive molecules from the OB [91,92] and chemorepellent molecules from the septum and choroid plexus [93,94,95]. The chains of migrating neuroblasts in the RMS are enwrapped by astrocytes which separate them from the mature tissue of the brain. It was initially believed that astrocytic tubes represent a physical barrier restricting this migration, and a scaffold navigating neuroblasts toward the OB [5]. However, it was subsequently observed that astrocytes do not provide continuous wrapping around the migrating neuroblasts [17,18,44]. Moreover, astrocytes are not present in the RMS until the second postnatal week [54]. Logically, the question arises—how are the SVZ-derived neuroblasts navigated for such a distance along the migratory pathway toward the OB? Besides the astrocytic scaffold, blood vessels “emerged” as a potential candidate which could serve as a support for migrating neuroblasts. It was noticed that specifically-arranged blood vessels in the RMS are in striking contrast with the arrangement of blood vessels in the surrounding brain parenchyma [13,16,17,18].

### 4.2. Region-Dependent Distinctions Relating to Blood Vessels in the RMS

The RMS in rodents is several millimeters long (approximately 5 mm in mice) [3] and is sigmoidal in shape. For the purposes of morphological analysis, the RMS is usually divided into smaller anatomical parts, however the terminology of these parts is not consistent among authors. It is possible to distinguish the following anatomical regions along the caudal–rostral axis of the RMS: (I) the vertical arm, the segment descending from the SVZ, leading ventrally, which can be subdivided into the caudal part, also known as the anterior part of the SVZ (SVZa) and the rostral part, also known as the vertical limb or descent; (II) the elbow: the most ventral segment of the RMS, which turns rostrally; (III) the horizontal arm, also known as the horizontal limb or enter, the segment leading rostrally, the horizontal arm, also known as the horizontal limb or enter, the segment leading rostrally, entering the OB (Figure 1) [3,17,18,37].

Although exact borders between the RMS regions cannot be recognized, some differences relating to blood vessels in these regions are obvious. For example, different blood vessel density in individual regions of the RMS in both, adult mice [17] and rats [18] and different arrangement of blood vessels along the caudo-rostral axis of the RMS in rats (see below) [18]. Moreover, differing permeability of blood vessels has been found along the SVZ-RMS-OB axis, with highly-permeable blood vessels in the SVZ and impermeable blood vessels in the RMS/OB [13,96]. Even though the reasons for these region-dependent distinctions remain unclear, one of the possible explanations could be the different developmental base of individual parts of the RMS. According to Pencea and Luskin [37], the RMS arises in advance and independently of the cortical SVZ and is derived from two sets of cells with different mitotic activities called patch and nonpatch. Although patch and nonpatch regions merge postnatally, they may originate separately under the influence of distinct intrinsic and extrinsic factors [37].

### 4.3. Specific Arrangement of Blood Vessels in the Rodent RMS and Interspecies Differences 

The arrangement of the RMS blood vessels was first described in adult mice [16,17]. According to these studies, blood vessels in adult mice precisely outline the migratory stream from its posterior (SVZ) to the most anterior (OB) regions [16], and their orientation is parallel to the RMS throughout its extent [17]. According to these studies, blood vessels in adult mice precisely outline the migratory stream from its posterior (SVZ) to the most anterior (OB) regions [16], and their orientation is parallel to the RMS throughout its extent [17]. In a later morphological study, we showed that this parallel arrangement of blood vessels is not applied to the whole RMS in rats [18]. We found clear differences in the arrangement of blood vessels between posterior and anterior regions of the RMS. In the posterior part of the rat RMS (the caudal part of the RMS vertical arm, lying under the corpus callosum), blood vessels are oriented perpendicular to the migratory pathway, or they are turned under a distinct angle, creating a spiral-shaped configuration (Figure 2A). In contrast to this, in the rest of the migratory pathway, i.e., in the rostral part of the vertical arm as well as the elbow and the horizontal arm of the RMS in adult rats, the orientation of blood vessels is parallel to the migratory stream (Figure 2A) [18]. Our preliminary results comparing the pattern of the RMS blood vessels in adult mice and rats revealed that there were not so striking differences in the arrangement of blood vessels between the species, whereby even in mice, the blood vessels are not aligned parallel to the migratory pathway, at least in its most posterior part (SVZa) (Figure 2B) (unpublished data). A detailed morphological study is needed to clearly show the interspecies differences in blood vessel arrangement. On the other hand, the pattern of blood vessel arrangement in the RMS of both species, mice and rats, is easily distinguishable from the rather random arrangement of blood vessels in the surrounding brain parenchyma (Figure 2A,B).

Nevertheless, there are some interspecies differences in the RMS of mice and rats, which are unrelated to blood vessels. These differences concern the RMS morphology and cellular composition. In parasagittal brain sections, the RMS has an “L” shape in adult rats (Figure 2A) but a “U” shape in adult mice (Figure 2B) [97]. A slight temporal delay can be observed in mice compared with rats regarding the glial structure assembly [44]. Furthermore, interspecies differences have been found in the molecular profile of certain glial cell populations in the SVZ-RMS. In addition, the first glial fibrillary acidic protein (GFAP) positive cells appear in the SVZ at P9 in rats and at P13 in mice [44]. The olfactory ventricle surrounded by the RMS and communicating with the lateral ventricle disappears at P3 in mice and at P6 in rats. Neuroblasts arranged in chains appear in mice at P15 and in rats at P21 [44]. Such differences are probably linked with the evolutionary divergence of mice and rats.

### 4.4. Blood Vessels, Neuroblasts, and Astrocytes in the RMS in the Context of Migration

Besides the specific arrangement of blood vessels in the RMS, another characteristic is their higher density in the migratory pathway in comparison with adjacent tissue of the brain in both species, mice, and rats [17,18]. Higher density of blood vessels applies to all parts of the RMS, and together with the known high density of cells in the RMS, this could logically suggest higher metabolic demand in this area. However, high blood vessel density may not be causally related to high cell density, since Whitman et al. [17] showed that the granule cell layer of the cerebellum, an area as equally cell-dense as the RMS, has a lower density of blood vessels. Recently, it has been shown that the higher density of blood vessels, as well as their specific arrangement, plays an important role in the RMS. Besides supplying nutrients, oxygen, and regulatory cues and being a part of the blood-brain barrier, another role of blood vessels in the neurogenic area of the adult brain has been proposed linked to neuroblast migration. Several studies have shown that there is evident association between blood vessels and the chains of neuroblasts. These studies suggest that blood vessels together with astrocytes might serve as a physical substrate fostering neuroblast migration from the SVZ to the OB [16,17,18,87]. The first indication of vasculature-guided (vasophilic) neuroblast migration appeared in the study of Bovetti et al. [87] and concerns the migration of neuronal precursors in the OB of mice. Other studies provided further evidence for the vasophilic mode of neuroblast migration in the RMS of adult mice [16,17]. Whitman et al. [17] found that approximately 50–85% of the length of each vessel is lined with chains of migrating neuroblasts, and this linear apposition increases along the course of the RMS. Quantitative analysis, measuring the distance of neuroblasts from blood vessels, revealed that in the RMS, around 97% of neuroblasts are located within 3 µm of the blood vessels [16]. In the part of the RMS entering the OB, where neuroblasts change their mode of migration from tangential to radial, this percentage was lower because some neuroblasts detach from blood vessels [16]. However, even in the granular cell layer of the OB (the final destination for the majority of new-born neurons), 45% of neuroblasts are distributed less than 3 µm from the blood vessels [87]. Time-lapse video imaging in acute brain slices of adult mice showed neuronal precursors migrating tangentially along the blood vessels in the RMS [16] and radially in the granular cell layer of the OB [87]. It was revealed that the neuroblasts migrated in a saltatory fashion along the blood vessels. This movement consisted of a migratory phase in which the neuronal cell body shifted toward the leading process, which was then followed by a stationary phase [16,87].

In the RMS, the chains of migrating neuroblasts are surrounded by GFAP-expressing astrocytes. It has been shown that the morphology of astrocytes in the RMS differs from that in the surrounding tissue. The astrocytes in the RMS of adult mice do not exhibit the typical stellate shape; they are rather polarized, and their branches are aligned with the path of migration [17]. Similarly, in adult rats, the majority of astrocytes have their long processes aligned with the migratory pathway; however, in the caudal part of the vertical arm (overlaid by the corpus callosum), most astrocytic processes are directed toward the corpus callosum, irrespective of the direction of neuroblast migration [18]. The astrocytes of the RMS line the blood vessels, and their processes interdigitate with the neuroblasts. It has been found that migrating neuroblasts can actively remodel astrocytic tubes to facilitate their directed migration. The neuroblasts have the ability to remove impeding astrocytic processes by secreting a diffusible protein with repulsive activity, Slit1, whose receptor, Robo, is expressed on the RMS astrocytes [98]. Astrocytes wrap the brain vasculature and usually separate neural cells from endothelial cells, contributing thus to the blood–brain barrier. Intriguingly, based on electron-microscopic observations, it has been demonstrated that neuroblasts in the RMS can in some cases directly contact blood vessels without any intervening astrocytes or pericytes [17,44]. Similarly in the SVZ, Tavazoie et al. [13] found that stem cells and transit-amplifying cells directly contact blood vessels at sites devoid of astrocyte endfeet, as well as pericyte coverage. Unlike the SVZ and RMS, such direct contact between neural precursor cells and endothelial cells has not been found in the OB [87]. 

### 4.5. Migration-Promoting Function of the Blood Vessels in the RMS

Recently, there has been a growing body of evidence concerning the vasophilic migration of neuroblasts in the RMS and the OB. However, it was still unclear whether blood vessels represent the “path of least resistance” for neuroblasts or whether blood vessels are also a source of some migration-promoting factors. Based on tracer experiments, it was found that the blood–brain barrier of the SVZ is modified and that circulating small molecules can enter the SVZ via two routes, directly from the vessels in the SVZ vascular plexus and from vessels in the choroid plexus [13,97]. Neuronal precursors residing in the SVZ may, therefore, easily receive spatial cues and regulatory signals circulating in the bloodstream. Moreover, endothelial cells themselves produce diffusible signals, such as VEGF, BDNF and other factors, which can affect neuronal precursors [79,99,100,101]. The regulatory effect of BDNF on migration of neuroblasts in the SVZ-RMS has been shown in vitro [102]. Later, the Saghatelyan group demonstrated that in the SVZ-RMS, BDNF is produced by endothelial cells of blood vessels [16]. Moreover, they revealed the principle of vasophilic migration [16,103] and they showed that migrating neuroblasts control their own migration by regulating the amount of extracellular BDNF. BDNF produced by endothelial cells bound with p75NTR low-affinity receptors on neuroblasts, thus fostering the entrance of neuroblasts to the migratory phase. Then, GABA released by neuroblasts induced Ca^2+^-dependent expression of TrkB, high-affinity receptor for BDNF, on astrocytes. This resulted in trapping of BDNF by astrocytes, thereby regulating its availability in extracellular space, which induces entry of migrating neuroblasts to the stationary phase [16].

As well as in the adult brain, vasophilic migration of neuroblasts has been observed in neonates. In neonatal and juvenile mice, besides tangential migration of neuroblasts along the RMS, radial migration of neuroblasts from the SVZ-RMS through the corpus callosum into the lower layers of the brain cortex has been observed [104]. Later, it was found that these neuroblasts migrate along the blood vessels and that the vasophilic mode of radial migration toward the cortex gradually decreases and disappears around the fourth postnatal week [105]. It was suggested that the decrease in vasophilic radial migration was probably due to the establishment of a glial sheath surrounding the RMS, and due to gradual decrease in the density of blood vessels in the corpus callosum [105]. However, tangential vasophilic migration of SVZ-derived neuroblasts toward the OB continues even in adulthood [16,17,87].

### 4.6. Development of a Vascular Scaffold in the RMS during Perinatal and Early Postnatal Periods

During perinatal and early postnatal developmental stages, intense vascular remodeling takes place in the forebrain neurogenic region [67,68]. At early postnatal stages (during the first postnatal week), blood vessels become more aligned in the longitudinal direction of the RMS and parallel to each other [68]. During later postnatal weeks, blood vessels continue increasing in complexity and length [67]. Mechanisms leading to the appearance and formation of a migration-promoting vasculature scaffold during early developmental stages were examined by Bozoyan et al. [20]. During early postnatal stages, the RMS contains two distinct sub-regions, the border and the core, with different characteristics [20]. Overall density of blood vessels in the RMS increases during postnatal development, with consistently more parallel blood vessels at the border of the RMS than in the core. Other differences have been found between the borders and the core of the RMS: fewer branched blood vessels in the borders than in the core; the presence of proliferating and migrating endothelial cells during early developmental stages restricted to the outer borders of the RMS, the place where the first parallel blood vessels appear. Thus, it has been suggested that the migration-promoting vasculature scaffold is first laid down on the borders of the RMS [20]. As the brain grows, the RMS gradually becomes longer and thinner, the center of the RMS collapses, and the RMS borders approach each other [20].

### 4.7. Relevance of Blood-Vessel Reorganization during the Early Postnatal Period for Migration of Neuroblasts

Although the RMS blood vessels are laid down during early development [67], their reorganization occurs postnatally [20] and this vascular rearrangement seems to be crucial for neurogenic processes in the SVZ-RMS-OB. By blocking VEGF signaling during early developmental stages, Licht et al. [106] induced a collapse of the RMS vascular network, which resulted in complete failure of the migration of newborn neuronal precursors from the SVZ toward the OB and thus piling up of neuronal precursors amidst the RMS and their subsequent apoptosis. In contrast, when VEGF signaling was blocked in the adult brain, no excessive endothelial or neuronal cell death was noted, and nor was any reduction seen in the number of neuronal precursors reaching the OB [106]. Later, it was shown that during the early postnatal period, VEGF is produced by astrocytes, which first emerge on the outer borders of the migratory stream and this astrocyte-derived VEGF plays a key role in the development and structural rearrangement of the vasculature scaffold [20]. When the VEGF expression was downregulated in vivo, specifically in the astrocytes of the developing RMS, the development of the vascular scaffold was affected and this consequently led to alteration of the neuronal migration, accumulation of neuronal precursors in the RMS, and decrease in the number of newborn neurons arriving at the OB [20]. In our laboratory, we obtained similar results when angiogenesis was inhibited during the early postnatal period [21]. Administration of endostatin, an endogenous inhibitor of angiogenesis which interferes with VEGFR-2, prevented the effect of VEGF on angiogenesis and resulted in the disruption of blood vessel reorganization into a proper vascular scaffold [21]. This failure of vascular rearrangement caused disruption of the mode and direction of neuroblast migration. Chain migration failed, and some neuroblasts moved individually and migrated out of the RMS. The number of proliferating cells in the RMS increased, probably due to their accumulation caused by disrupted neuroblast migration. These effects of angiogenesis inhibition were more evident shortly after the inhibition, although the effects remained noticeable even after a longer time [21]. Taken together, this suggests that the reorganization of blood vessels in the RMS and formation of a vascular scaffold during the early postnatal period is crucial for the regular course of postnatal neurogenesis in the SVZ-RMS-OB system.

## 5. Conclusions

Neurogenesis in the adult brain attracts the attention of researchers all over the world because there are prospects for the use of newly-generated cells in treating disorders associated with neuron loss, such as brain injury, neurodegenerative diseases, or stroke. Under physiological conditions, newborn cells originating in the SVZ are destined for the OB. In addition, under pathological conditions, neuronal precursors are able to migrate out of the neurogenic region of the SVZ toward the site of ischemia [72,107]. Adult neurogenesis is a complex process, which includes proliferation, migration, differentiation, and integration of newborn cells into functional neuronal circuits. Each of these processes is regulated by multiple factors on cellular and molecular levels.

Although blood vessels have been known for many years to be a critical component of neurogenic niches [13,14,74,76], it has only recently been revealed that the vasculature in the SVZ-RMS-OB also plays a role in the regulation of neuroblast migration [16,17,87]. Blood vessels in the adult neurogenic region serve as a scaffold along which neuroblasts migrate (Figure 3), similar to processes of radial glial cells during the embryonic period. In addition, endothelial cells of blood vessels produce migration-promoting cues [16]. 

In this review, recent findings on vasculature-guided neuroblast migration in the largest neurogenic region of the adult brain, the SVZ-RMS-OB, with an emphasis on comparison of two rodent species, mouse, and rats were summarized. We gathered knowledge on development of the telencephalon and its vascularization in the presumptive neurogenic region of rodents during the embryonic period, in an effort to better understand the relationship between the emerging vasculature in embryogenesis and the vasculature in the postnatal period. It is important to emphasize that not only is the laying down of blood vessels during the embryonic period necessary for the proper migration of neuroblasts to the target site, their reorganization into a migratory scaffold is also crucial. It is currently known that astrocytes regulate the reorganization of blood vessels into a migratory scaffold in the early postnatal period via VEGF signaling [20]. The blocking of VEGF signaling during this time frame leads to disruption of that specific blood vessel reorganization, and consequently disturbs neuroblast migration [20,21].

Although we currently have a wealth of knowledge in the field of postnatal neurogenesis, research is still ongoing because detailed knowledge of individual processes and regulation of neurogenesis may lead to a better understanding of the functional significance of adult neurogenesis, and possibly to the use of an endogenous source of new cells to replace damaged neurons in the future.

## Figures and Tables

**Figure 1 ijms-22-11506-f001:**
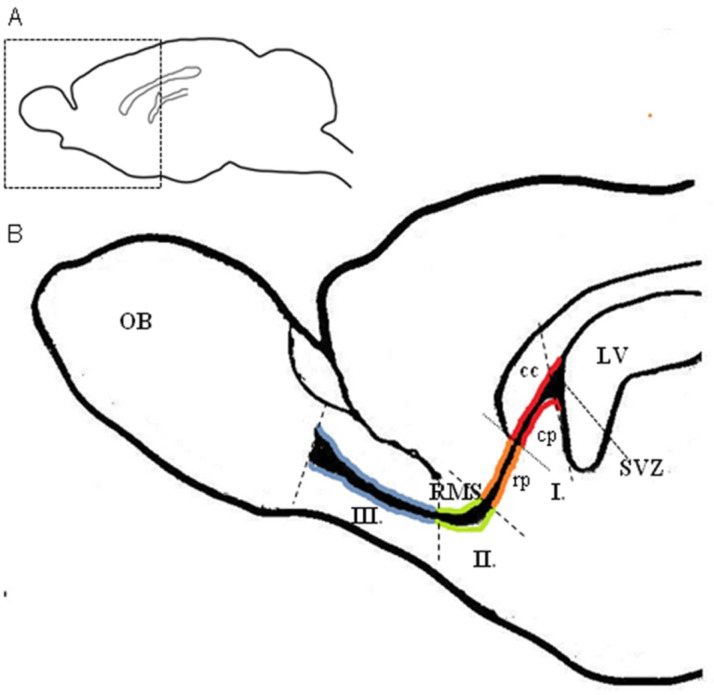
Anatomical regions of the RMS. (**A**) Schematic drawing of a sagittal section of the adult rat brain. (**B**) Schematic drawing of magnification of the boxed area of the picture (**A**) showing the division of the RMS into anatomical regions. The regions are highlighted with different colors. In the caudal to rostral direction, the following can be recognized: the vertical arm (I), which can be subdivided into the caudal part (cp—red) lying under the corpus callosum and the rostral part (rp—orange) leading ventrally; the elbow (II—green) and the horizontal arm (III—blue) leading rostrally toward the OB. LV—lateral ventricle, cc—corpus callosum, OB—olfactory bulb, RMS—rostral migratory stream, SVZ—subventricular zone.

**Figure 2 ijms-22-11506-f002:**
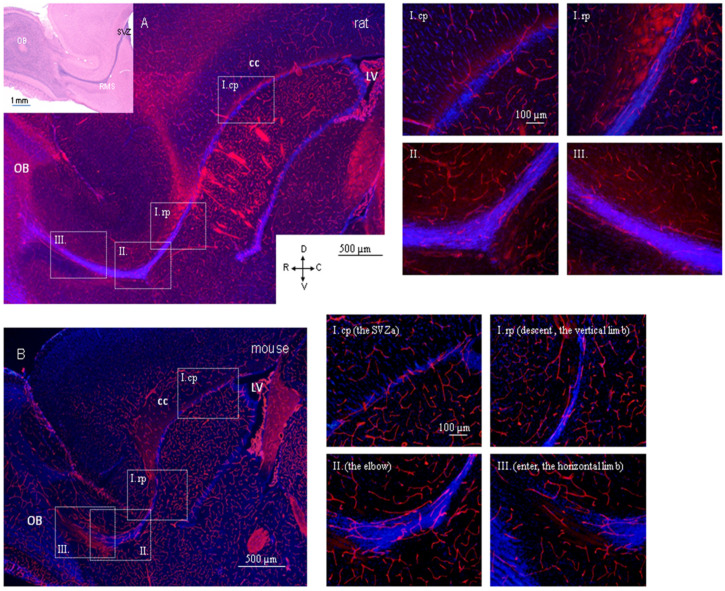
Arrangement of blood vessels in the RMS of adult rodents. Micrographs show organization of PECAM-1-labelled blood vessels (red) in the RMS of adult rat (Wistar albino) (**A**) and mice (Balb/c) (**B**). Nuclei were counterstained with DAPI (blue). Inset in A shows sagittal section of the rat brain processed with haematoxylin-eosin staining. The RMS is visible as an L shape strip of densely-packed cells. I—vertical arm of the RMS, cp—caudal part of the vertical arm, rp—rostral part of the vertical arm, II–elbow of the RMS, III—horizontal arm of the RMS, OB—olfactory bulb, cc—corpus callosum, LV—lateral ventricle, SVZ—subventricular zone, D—dorsal, V—ventral, C—caudal, R—rostral.

**Figure 3 ijms-22-11506-f003:**
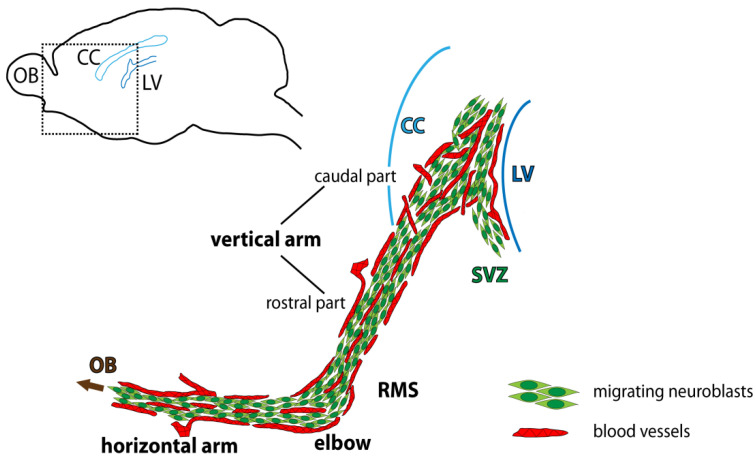
Vasculature-guided neuroblast migration in the RMS. Schematic drawing depicting neuroblasts migrating in chains along blood vessels. SVZ—subventricular zone, RMS—rostral migratory stream, OB—olfactory bulb, CC—corpus callosum, LV—lateral ventricle.

**Table 1 ijms-22-11506-t001:** Developmental milestones of telencephalon, OB, and RMS formation.

Developmental Event	Detail	Timing	Mice Strain/ Reference	Detail	Timing	Rat Strain/ Reference
Establishment of CNS	Formation of neural plate	E7–E7.5	CD1 (ICR) [22]	Neural plate stage	E9	Sprague-Dawley [23]
Neurulation	Initiation at hindbrain/ cervical boundary	E8.5	not specified [24]	Neural tube closed in upper thoracic and lower cervical region	by E10.5	Sprague-Dawley [23]
	Closure of the anterior neuropore	E9	normal hy-3 [25]	Closure of the anterior neuropore	E10.5	not specified [26]
	Completion of rostral neural tube—segmentation to primary brain vesicles	E9	CD1 (ICR) [22]	Entire neural tube is closed	by E11, by E12	Sprague-Dawley [23], Purdue-Wistar [27]
	Segmentation to secondary brain vesicles—development of telencephalic vesicles	E9.5	CD1 (ICR) [22]	Segmentation to primary brain vesicles	E11–E12	Purdue-Wistar [27], Fisher [28]
				Segmentation to secondary brain vesicles—development of telencephalic vesicles	E12–E13, E13	Fisher [28], Purdue-Wistar [27]
Morphogenesis of telencephalon, OB and RMS	Early ventral forebrain cells acquire general migratory capacity	E9.5–E11.5	ICR [29]	Telencephalon wall divides to VZ, IZ, MZ	E13–E15	Sprague-Dawley [30], Fisher [28], Wistar [31]
	Telencephalon wall divides to VZ, IZ, MZ	E10–E10.5	CD1 (ICR) [22]	Olfactory nerve fibers reach the telencephalon	E13, E14	Sprague-Dawley [30], not specified [32]
	Onset of mitral cells production	by E11, E10–E12	B6tgN [33], CD1 [34]	Olfactory nerve fibers penetrate the VZ of telencephalon	E14	Wistar [31]
	Morphogenesis of subpallial MGE	E11–E12	ICR [35], ICR [29], not specified [36]	Development of neurogenic anterior SVZ	from E14	Sprague-Dawley [37]
	Morphogenesis of subpallial LGE	E12–E12.5	ICR [29], not specified [36]	Differentiation of neuronal cells of prospective RMS	E14	Sprague-Dawley [37]
	Generation of subpopulation of OB projection neurons in the rostral LGE of cultured embryo	E11–E12	C57 mice [38]	Mitral cells accumulate at the base of telencephalon	E15	Purdue-Wistar [27], Sprague-Dawley [30], Sprague-Dawley [37]
	Generation of main population of mitral cells	E11–E13	CD1 [34], F1 hybrids of Balb/c females and SJL/J males [39], C57 BL/6 [40]	peak of mitral cells generation	E14–E16	Purdue-Wistar [41]
	LGE cells acquire migratory capacity toward the OB	E11.5–E12.5	ICR [29]	OB evaginations emerge in anterior telencephalon	E15	Purdue-Wistar [27], Sprague-Dawley [30], Sprague-Dawley [37]
	Primordial OB detectable at the anterior tip of telencephalon	E12.5	not specified [42]	Neuronal progenitors organize into prospective RMS	by E15	Sprague-Dawley [37]
	Radial glial cells are present in the developing OB	E13.5	not specified [43]	Neuronal cells organize into dense compact patch; GFAP positive cells emerge in forming RMS	by E16	Sprague-Dawley [37]
	Main production of tufted cells	E14–E17	F1 hybrids of Balb/c females and SJL/J males [39]	Peak of internal tufted cells generation	E16–E17	Purdue-Wistar [41]
	Generation of neuronal populations of presumptive RMS originating in LGE	E14.5	ICR [29]	RMS emerges in the rostral forebrain; neuronal patch is surrounded by non-patch cells	by E17	not specified [32], Sprague-Dawley [37]
	Presence of olfactory lobes at the rostral end of telencephalon	E15–E15.5	CD1 (ICR) [22]	OB contain evaginated parts of the lateral ventricles	E18	Purdue-Wistar [27]
	Organization of neuronal cells into presumptive RMS	by E16.5	ICR [29]	Peak of generation of tufted cells: external interstitial	E18–E19 E20–E22	Purdue-Wistar [41]
	Shrinkage of lateral ventricles, rostral extensions into the OB	E16.5–E18.5	CD1 (ICR) [22]	Peak of generation of periglomerular cells and external plexiform layer cells	P0–P7	Purdue-Wistar [41]
	Main production of OB granule cells	E18–P20	F1 hybrids of Balb/c females and SJL/J males [39]	Peak of production of granule cells (continues throughout the rest of life)	P0–P15	Purdue-Wistar [41]
	OB ventricles: closed still present	P0–P1 P0–P3	not specified [43], CD1 [44]	OB ventricles: still present	P3–P4	Wistar [44]
	Expression of GFAP in astrocytes	P6–P13	CD1 [44]	Expression of GFAP in astrocytes	P6–P9	Wistar [44]
	glial tubes emerge in the RMS	P21–P25	CD1 [44]	Glial tubes emerge in the RMS	P21–P25	Wistar [44]

CNS = central nervous system; E = embryonic day; P = postnatal day; VZ = ventricular zone; IZ = intermediate zone; MZ = marginal zone; MGE = medial ganglionic eminence; LGE = lateral ganglionic eminence; OB = olfactory bulbs; RMS = rostral migratory stream; BV = blood vessels; GFAP = glial fibrilary acidic protein.

**Table 2 ijms-22-11506-t002:** Vascularization of telencephalon and RMS.

Developmental Event	Detail	Timing	Mice Strain/ Reference	Detail	Timing	Rat Strain/ Reference
Vascularization of telencephalon: Vasculogenesis	Initiation of PNVP in the mesoderm surrounding the neural tube	E8.5	Swiss albino [60]	PNVP covers lateroventral surface of the rostral neural tube	E11	Sprague-Dawley [56]
	PNVP covers the surface of telencephalon	E9	CD1 [63]	Degeneration of capillaries in the meningeal plexus	after E15	Sprague-Dawley [56]
	Region over telencephalic roof plate remains devoid of PNVP	E8.5–E9.5	Swiss albino [60]	end of leptomeningeal vasculature sprouting into the cerebral cortex	P8–P15	Sprague-Dawley/Wistar [69]
	Whole neural tube is covered by PNVP	E11.5, E10–E12	Swiss albino [60], normal hy-3 [25]			
	Superficial vasculature condenses to tubular vessels	by E12	normal hy-3 [25]			
Vascularization of telencephalon: Angiogenesis	Formation of PVVP in ventral telencephalon	E9–E10	CD1 [63]	Onset of internal vascularization of telencephalon	E13	Fisher [28]
	Progression of PVVP from ventral to dorsal telencephalon	E11	CD1 [63]	Formation of deep vascular plexus of the VZ (PVVP)	E13	not specified [64]
	PVVP emerges in dorsal subpallium and in ventral pallium	E11.5	Swiss albino [60]	Decrease in proliferation of endothelial cells in telencephalon	after P20	Sprague-Dawley/Wistar [69]
	Simple loops of vascular plexi surround the rostral extension of lateral ventricles	E14	C57Bl/6 [68]			
	Early phase of angiogenesis in dorsal telencephalon (sprouting, branching)	E14.5–E16.5	C57BL/6 [65]			
	Late phase of angiogenesis in the dorsal telencephalon (fusion of branches)	E16.5–E18.5	C57BL/6 [65]			
	Cease of angiogenesis in mice brain	after P20	not specified [70]			
Vascularization of neurogenic region of forebrain	presence of short, straight, unbranched BV	E14.5	CD1 [67]			
	BV are tangentially oriented in the presumptive RMS	E16	C57BL/6 [68]			
	BV are longitudinaly organized in the RMS—parallel to each other	by E16–P4	C57Bl/6 [68]			
	BV are longer, branched, tangentially oriented, follow longitudinal axis of forming RMS, more frequent along the border of RMS	E17.5	CD1 [67]			

E = embryonic day; P = postnatal day; VZ = ventricular zone; PNVP = perineural vascular plexus; PVVP = periventricular vascular plexus; RMS = rostral migratory stream; BV = blood vessels.

## Data Availability

The data presented in this study are available upon request.

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
