# Peer review of "Relationship between Blood Vessels and Migration of Neuroblasts in the Olfactory Neurogenic Region of the Rodent Brain"

_ijms, 2021, doi:10.3390/ijms222111506_

Round 1

Reviewer 1 Report

The review article by Marcela Martončíková and colleagues addresses an important issue concerning the role of blood vessels in relation to neuroblast migration in the subventricular  zone (SVZ) - the rostral migratory stream (RMS) - the olfactory bulb (OB), the development  and  vascularization  of  the  presumptive  neurogenic region during the embryonic period, the relevance of blood vessel rearrangement in the RMS during the early postnatal development and other aspects of the phenomenon. Overall this review article is interesting and important in the context of neurogenesis in  the adult  brain. Also, the number of references cited is significant and the reference list covers the relevant literature adequately.

However, the manuscript has several shortcomings. My major criticisms are as follow:

  1. The Authors, before they started to write this article, should have been decided whether the narrative or systematic review is prepared. The current form of the manuscript is a mixture of both kinds and it should be revised to meet the criteria of one of the two types of review articles (either a narrative or systematic review).
  2. The Table 1 is poorly readable as it is currently 7 pages long. Please reorganize this table in a way that improves its readability, e.g., it can be divided into several smaller tables, the page orientation can be changed from portrait to landscape.
  3. In the conclusions section, a paragraph should be added that will emphasize what new this article brings in comparison to other earlier articles on the topic.

Author Response

Dear reviewer,

We would like to thank you for your fruitful comments, which we appreciate very much. In accordance with your comments, a revision of the MS was done and all remarks, raised by you, were considered.

Point 1: The Authors, before they started to write this article, should have been decided whether the narrative or systematic review is prepared. The current form of the manuscript is a mixture of both kinds and it should be revised to meet the criteria of one of the two types of review articles (either a narrative or systematic review).

Response 1: We agree with you regarding the presence of two different styles in the manuscript. Due to the vastness of the literature, we divided the individual parts among the authors. The second reason is that we do not have our own results in embryonic development research, so we could provide only systematic review. However, in light of the findings about vasculature-guided neuroblast migration in the neurogenic region in the postnatal period, we considered it important to include embryonic development of these areas of the brain and their vascularization as well. When revising the manuscript, we tried to change the style of systematic review so that the differences were not so sharp.

Point 2: The Table 1 is poorly readable as it is currently 7 pages long. Please reorganize this table in a way that improves its readability, e.g., it can be divided into several smaller tables, the page orientation can be changed from portrait to landscape.

Response 2: We accepted your comment and reorganized the Table 1: we changed the page orientation to landscape and divided the table into two tables, (Please note that dividing into several smaller tables would disrupt the structure of the whole chapter) Tab. 1 is devoted to the development of telencephalon, OB and RMS, Tab. 2 is devoted to the vascularization of the telencephalon and RMS. We hope these adjustments increase the readability.

Point 3: In the conclusions section, a paragraph should be added that will emphasize what new this article brings in comparison to other earlier articles on the topic.

Response 3: In the past years, significant progress has been made to support the role of blood vessels in neuroblasts migration in the olfactory neurogenic region in postnatal period.  So, there emerged a need to summarize knowledge on this topic, which is the purpose of this review. Our effort was also to provide information about the differences between the most commonly used model animals, mice and rats and to gather knowledge on development of the telencephalon and its vascularization in the presumtive neurogenic region of rodents during the embryonic period, in an effort to better understand the relationship between vasculature in embryogenesis and the vasculature in the postnatal period.

These facts were added to the Conclusion section of the revised manuscript.

Reviewer 2 Report

ijms-1408605: Relationship between blood vessels and migration of neuroblasts in the olfactory neurogenic region of the rodent brain

This review is one of the best recent ones about the neural development research field. The summary table and figures are excellent and useful. Those who intend to start the research of this field should read this review. This review is also useful to the experts of this field to cover the current research status. Especially, the Table 1 may be a very useful dictionary to study the development of rat or mouse brain. The manuscript is almost complete and can be published in the present form. As some points, which may improve the value of this review a little bit, are suggested as follows, the decision is "accept after minor revision". However, this manuscript is almost "accept in present form."

<Minor points>
(1) In the conclusion section, a summary figure may be included to depict the relationship between blood vessels and migration of neuroblasts in the olfactory neurogenic region at a glance.
(2) In the Abstract, it should be included that this review mainly describes the works of rats or mice. For example, line 20: Here we review <the studies of the rodent brain about> the development of vasculature in the presumptive neurogenic region...
(3) The summary section of the differences between rats and mice should be included. In each section, the results of rats and mice are described almost in parallel. An additional section with a graphical summary before the last conclusion section is recommended to summarize the difference between rats and mice. This section may also include the brief description of the difference between human and rodents for the medical research.
(4) In Fig.1 the figure of the whole brain with low magnification may be added to show the orientation of the olfactory bulb for the beginners. To understand Fig. 2 more clearly, the picture of haematoxylin and eosin staining may be also recommended.
(5) The list of the abbreviations used may be convenient.
(6) The postal address of the authors should be included. Although the e-mail address of the authors may be enough for communication, the complete postal address of the authors should be specified in accordance with the tradition of scientific papers.

End of File

Author Response

Dear reviewer,

We would like to thank you for the comments concerning our manuscript. We carefully addressed all the comments and according to the advice changes were included in the revised manuscript.

Point 1: In the conclusion section, a summary figure may be included to depict the relationship between blood vessels and migration of neuroblasts in the olfactory neurogenic region at a glance.

Response 1: We accepted your comment. The summary figure illustrating the relationship between blood vessels and migrating neuroblasts is included in the conclusion section of the revised manuscript.

Point 2: In the Abstract, it should be included that this review mainly describes the works of rats or mice. For example, line 20: Here we review <the studies of the rodent brain about> the development of vasculature in the presumptive neurogenic region...

Response 2: In accordance with your request, the information that the review mainly describes the works on rodents is included in the abstract of the revised manuscript.

Point 3: The summary section of the differences between rats and mice should be included. In each section, the results of rats and mice are described almost in parallel. An additional section with a graphical summary before the last conclusion section is recommended to summarize the difference between rats and mice. This section may also include the brief description of the difference between human and rodents for the medical research.

Response 3: Since the differences in the arrangement of vessels in the neurogenic region of mice and rats are not as significant as originally thought, we do not have enough data to make a graphical summary. Some differences in the RMS between mice and rats unrelated to blood vessels are mentioned in the second paragraph of the chapter 4.3. Some specific distinctions between mice and rats in the development of the telecephalon and its vascularization are summarized in the Table 1 and Table 2, respectively (the original table was divided into two smaller ones on the request of the reviewer 2).

To our knowledge, the arrangement of blood vessels in the human olfactory neurogenic region has not yet been described, so we cannot compare this with rodents. However, the role of blood vessels in neuroblast migration is strongly suggested by findings about stroke-induced compensatory neurogenesis in the human brain. In patients with stroke, cells that express markers associated with newborn neurons are present in the ischemic penumbra surrounding cerebral cortical infarcts, where these cells are preferentially localized in the in the vicinity of blood vessels.

Point 4: In Fig.1 the figure of the whole brain with low magnification may be added to show the orientation of the olfactory bulb for the beginners. To understand Fig. 2 more clearly, the picture of haematoxylin and eosin staining may be also recommended.

Response 4: In accordance with your recommendation, the figure of the whole brain with low magnification was added to the Figure 1 and the picture of hematoxylin eosin staining was added to the Figure 2.

Point 5: The list of the abbreviations used may be convenient.

Response 5: The list of the abbreviations was added.

Point 6: The postal address of the authors should be included. Although the e-mail address of the authors may be enough for communication, the complete postal address of the authors should be specified in accordance with the tradition of scientific papers.

Response 6: We included the postal address of the authors.

Round 2

Reviewer 1 Report

The manuscript has been revised correctly. I have no further comments to the Authors.